# Ozone Induced Loss of Seed Protein Accumulation Is Larger in Soybean than in Wheat and Rice

**Malin C. Broberg \***, **Sara Daun and Håkan Pleijel**

Department of Biological and Environmental Sciences, University of Gothenburg, P.O. Box 461, SE-40530 Göteborg, Sweden; gusdaunsa@student.gu.se (S.D.); hakan.pleijel@bioenv.gu.se (H.P.)
\* Correspondence: malin.broberg@bioenv.gu.se; Tel.: +46-701-719-190

**Abstract:** We investigated the effects of ozone ($O_3$) on seed protein accumulation in soybean, rice, and wheat based on existing literature. We identified 30, 10, and 32 datasets meeting the requirements for soybean, rice, and wheat, respectively. Data for each crop were combined in response regressions for seed protein concentration, seed protein yield, and seed yield. Although seed yield in rice was less sensitive to $O_3$ than in wheat, there was a significant positive effect of $O_3$ on the seed protein concentration of the same magnitude in both crops. Soybean, an N-fixing high-protein crop, responded differently. Even though the effect on seed yield was similar to wheat, there was no indication of any effect of $O_3$ on seed protein concentration in soybean. The negative influence of $O_3$ on seed protein yield was statistically significant for soybean and wheat. The effect was larger for soybean (slope of response function: −0.58% per ppb $O_3$) than for wheat (slope: −0.44% per ppb) and especially compared to rice (slope: −0.08% per ppb). The different response of protein concentration in soybean, likely to be associated with adverse $O_3$ effects on N fixation, has large implications for global protein production because of the much higher absolute protein concentration in soybean.

**Keywords:** *Glycine max*; nitrogen; *Oryza sativa*; $O_3$; protein; response function; *Triticum aestivum*; yield

## 1. Introduction

Ground-level ozone ($O_3$) is formed in air masses with elevated levels of nitrogen oxides, volatile organic compounds (including methane), and carbon monoxide [1] under the influence of solar radiation. $O_3$ is known to negatively affect plant growth and consequently, the yield of several important crops [2,3], including wheat (*Triticum aestivum*) [4,5], soybean (*Glycine max*) [6,7], and rice (*Oryza sativa*) [8–10].

In addition to the loss of seed yield (SY), various effects on quality properties, such as nutrient concentrations and yield, have been observed in crops exposed to $O_3$. For example, positive effects on seed protein concentration (SPC), and reductions in unit area seed protein yield (SPY), have been observed consistently in wheat [11,12]. This can be explained by the stimulation of SPC by $O_3$ being smaller than the loss in SY, resulting in a net loss of protein accumulation as SPY [13]. In experiments where the rate of nitrogen (N) application is the same in the different treatments, the reduction in unit area SPY under $O_3$ exposure translates into a reduced N fertilizer efficiency [14], since for a certain crop, the protein-to-N ratio is more or less constant [15].

While the characteristics of the $O_3$ responses of protein accumulation and N fertilizer efficiency have been considered in several meta-analyses for wheat [4,12,16], corresponding investigations of $O_3$ impacts on protein accumulation in soybean and rice, both crops of major global significance, are scarce. The effects of $O_3$ on SPC in rice [17–19] and soybean [20–22] have, however, been reported for several experiments allowing for the synthesis of $O_3$ effects on SPY also for these crops.

Soybean is a highly important crop globally. It has symbiotic N fixation, typically resulting in much higher protein concentrations compared to cereals. Of the globally most important crops, soybean is among the most $O_3$ sensitive [2,23]. Since there is substantial evidence that N-fixation can be negatively affected by $O_3$ (reviewed by Hewitt et al. [24]), it is reasonable to hypothesize that any positive effects on SPC would be smaller and negative effects on SPY larger in soybean compared to cereals. The observations of Cheng et al. [25] showed that $O_3$ negatively affected N fixation in soybean in an open-top chamber experiment, which promotes such a hypothesis.

The $O_3$ sensitivity of rice considered the second most important crop of the world after maize, has been less studied than soybean and wheat. In general, it seems to be less sensitive to $O_3$ than wheat and soybean with respect to SY [2], still responding significantly to current or near-current levels of $O_3$ [10]. As already mentioned, SPC and SPY in wheat have been reported in many studies and were summarized by Broberg et al. [12]. It would be reasonable to assume, based on the smaller effect on seed yield in rice compared to wheat, that a similar response pattern of SPC enhancement and SPY loss under $O_3$ exposure would also apply to rice, but to a smaller extent than in wheat.

In this investigation, we compared the response of SPY and SPC to $O_3$ exposure of the three crops by combining available data from the experimental literature in response functions. Response functions were also derived for SY for comparison, but it should be noted that these functions did not include all available data for seed yield regarding the three crops, only those experiments reporting protein to allow direct comparison of the $O_3$ sensitivity of the three response variables. Our hypotheses were:

1.  SPC is less positively affected by $O_3$ in soybean compared to wheat and rice.
2.  SPY is more negatively affected by $O_3$ in soybean compared to wheat and rice, resulting from the high sensitivity of SY in combination with the first hypothesis.
3.  The absolute loss of protein production per unit area from $O_3$ is much larger in soybean compared to wheat and rice because of higher $O_3$ sensitivity in combination with a (much) higher protein concentration in soybean.

## 2. Materials and Methods

Web of Science, Scopus, and Google Scholar were used to search all peer-reviewed literature published between 1980 and 2019 (October) for experiments with soybean, rice, and wheat grown under manipulated $O_3$ exposure with a duration of >4 weeks. Experiments with field-grown crops or pot-grown crops under field conditions were included where data on (i) SY and SPC, and (ii) daytime mean $O_3$ concentration, $[O_3]_{day}$, were reported. SPY was calculated as the product of SY and SPC if it was not explicitly reported in the papers. We found 30, 10, and 32 datasets that could be included for soybean, rice, and wheat, respectively. Wheat and soybean experiments were from Asia, Europe, and North America, while the rice experiments were all conducted in Asia. The references from which the data were extracted, as well as the extracted data, are provided in Supplementary Materials (sup. With the exception of one investigation [26] (using $O_3$ monitoring based on chemiluminescence), $O_3$ monitoring was based on the UV absorption principle in all experiments.

The values of the response variable for a treatment in a particular experiment were related to the value estimated for zero $[O_3]_{day}$ using linear response extrapolation [27]. At zero exposure, the response variable was set to take the value of 1 on a relative scale, i.e., it was assumed that there was no $O_3$ effect associated with zero exposure in each experiment for each of the three response variables. Experiments with only two observations and an $[O_3]_{day}$ range (difference between highest and lowest $O_3$ treatment) smaller than 15 ppb were excluded to avoid the uncertainty in the determination of the intercept and the associated response extrapolation since random effects become large in relation to the difference in exposure for such experiments. After the transformation to a relative scale, the data from all experiments were combined to derive exposure–response relationships. The methodology for the calculations, including the regression method to define relative yields for individual experiments, has been described in detail by Grünhage et al. [13]. Further, the relationship between relative values of SY and SPC obtained using this approach for the three crops was investigated.

Where seed N rather than protein concentration was reported, data were converted to protein concentration by multiplying the N concentration with the crop-specific conversion factors provided by Mosse [15]: 5.52 for soybean, 5.33 for wheat, and 5.17 for rice.

Outliers in the response regressions were identified by the Rout method [28]. Any outliers are marked with pink shade in the associated supplementary file (Supplementary file 1). To allow for a direct comparison of the three graphs representing SY, SPC, and SPY for each crop, experimental treatments containing an outlier for one of the three response variables were excluded from the calculation of regression statistics as well as for the other response variables (marked yellow Supplementary file 1). Data points thus excluded from the statistical analysis are included in the graphs showing the response regression using open symbols. Based on the identification of outliers, 8 out of 70, 6 out of 40, and 6 out of 95 observations were excluded from the statistical analysis for soybean, rice, and wheat, respectively.

To investigate the implications of our results with respect to loss of protein and yield from $O_3$, the estimated absolute SPY based on absolute SY data from the Food and Agricultural Organization of the United Nations (FAO) [29], absolute SPC from the experiments (used in this study), and relative SPY estimated from the response regressions are presented for the three levels of $[O_3]_{day}$: 10 ppb (~preindustrial), 37 ppb (present), and a hypothetical projected further elevated $[O_3]_{day}$ level of 60 ppb. The average $[O_3]_{day}$ of the non-filtered treatment (representing the current ambient air $[O_3]_{day}$) was 44 ppb for soybean, 38 ppb for rice, and 35 ppb for wheat with a combined average of 37 ppb. It is hard to know the preindustrial background $[O_3]$ with certainty since different methods were used to monitor $O_3$ in the 19th century and the first part of the 20th century. Based on existing evidence, it has been estimated to have been ~10 ppb [30,31]. Global average unit area yields for the period 2013–2017 of the three crops were extracted from the FAO database. Protein concentrations used in this analysis were based on the SPC levels observed in the experiments covered by the study. More precisely, an average for each crop was made of the absolute SPC associated with zero $O_3$ in the regression between SPC and $[O_3]_{day}$ for each individual experiment.

## 3. Results

### 3.1. Effects of $O_3$ on Seed Yield (SY)

SY was significantly and negatively affected by increasing $O_3$ in all three crops (Figure 1a–c). Although all three response regressions were strongly significant, the sensitivity to $O_3$, indicated by the regression slope, was smaller for rice (slope coefficient: −0.0018) than for soybean (slope coefficient: −0.0058) and wheat (slope coefficient: −0.0050). The most consistent relationship was obtained for wheat ($R^2 = 0.72$), while that for rice had the weakest ($R^2 = 0.32$), soybean being intermediate in this respect ($R^2 = 0.54$).

As mentioned above, it should be noted that the relationships for SY only contain the experiments also reporting protein to make possible the direct comparison of all figures in this paper for a certain crop by using exactly the same experimental data. Several further experiments reporting SY, but not SPC/SPY, exist for all three crops [2].

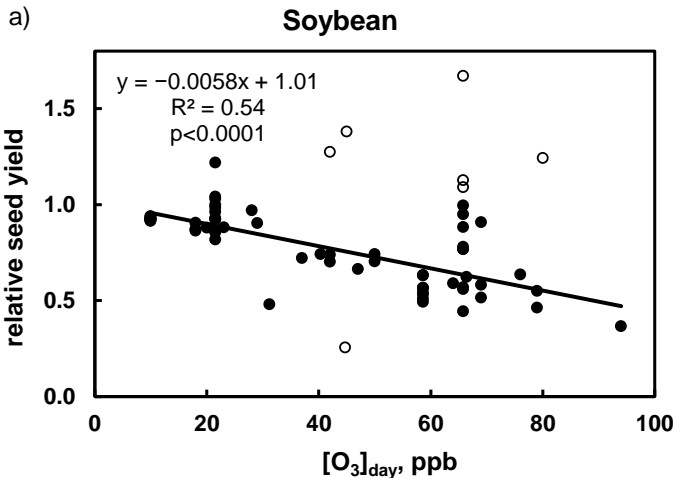

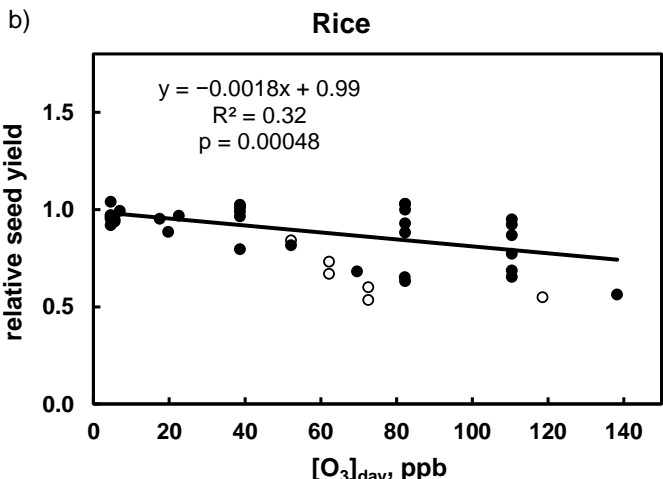

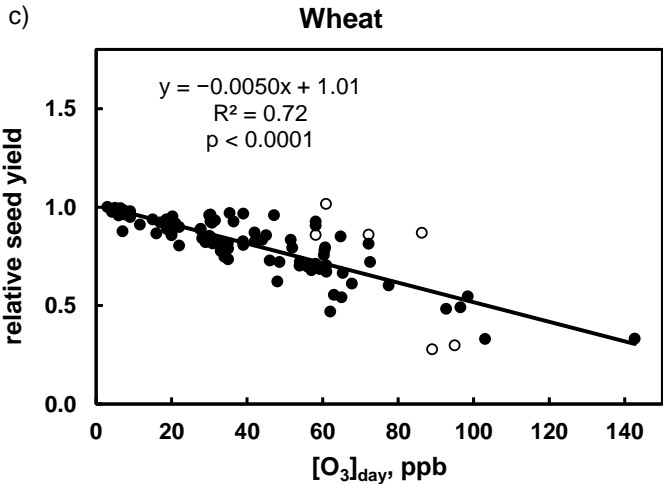

**Figure 1.** Relationship between relative seed yield (SY) and daytime $O_3$ concentration, $[O_3]_{day}$, for (**a**) soybean; (**b**) rice; (**c**) wheat. Open circles denote data points representing statistical outliers for at least one of the studied response variables.

### 3.2. Effects of $O_3$ on Seed Protein Concentration (SPC)

Unlike rice and wheat, there was no indication of a positive relationship between SPC and $[O_3]_{day}$ in soybean ($p = 0.32$), as can be inferred from Figure 2a–c. The absence of a response for this variable in soybean was highly consistent. The stimulation of seed protein concentration by $O_3$ in the two cereals, rice and wheat, were on the other hand, very similar with a slope coefficient of 0.0014 in both cases, and the response regressions for both crops were strongly statistically significant.

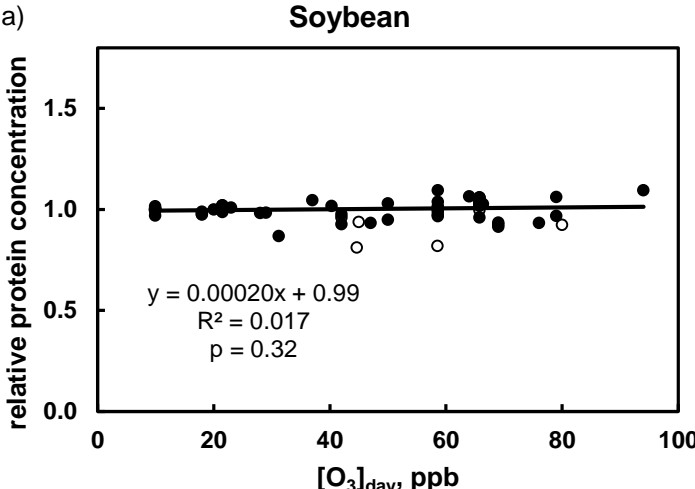

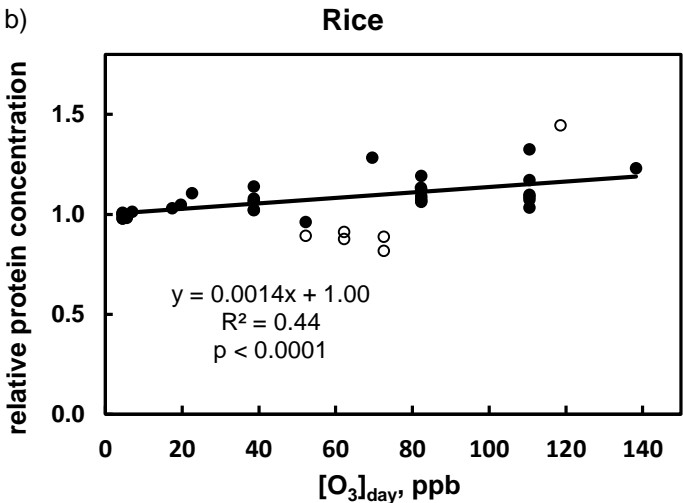

**Figure 2.** *Cont.*

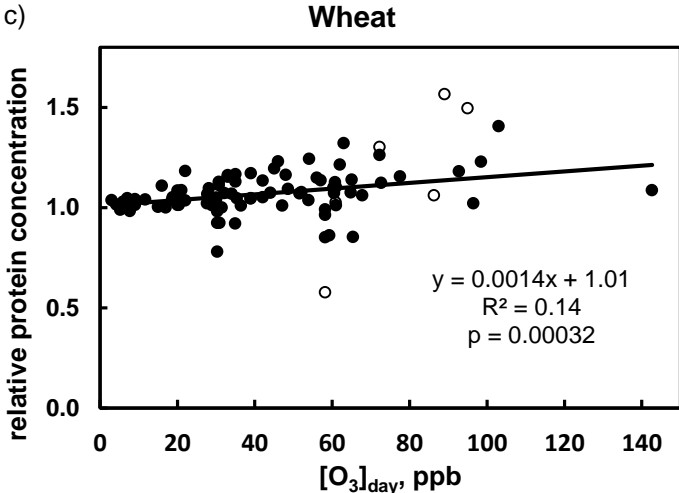

**Figure 2.** Relationship between relative seed protein concentration (SPC) and daytime $O_3$ concentration, $[O_3]_{day}$, for (**a**) soybean; (**b**) rice; (**c**) wheat. Open circles denote data points representing statistical outliers for at least one of the studied response variables.

### 3.3. Effects of $O_3$ on Seed Protein Yield (SPY)

As a consequence of the strong response of SY, and the absence of a positive effect on SPC, the negative effect of $O_3$ on SPY (slope coefficient: −0.0058) was highly significant ($p < 0.0001$) and larger in soybean than in the other two crops (Figure 3a–c). The sensitivity to $O_3$ of SPY was larger for wheat (slope coefficient: −0.0044) than for rice (slope coefficient: −0.0008). The response regression for wheat was strongly significant, but not in the case of rice ($p = 0.077$).

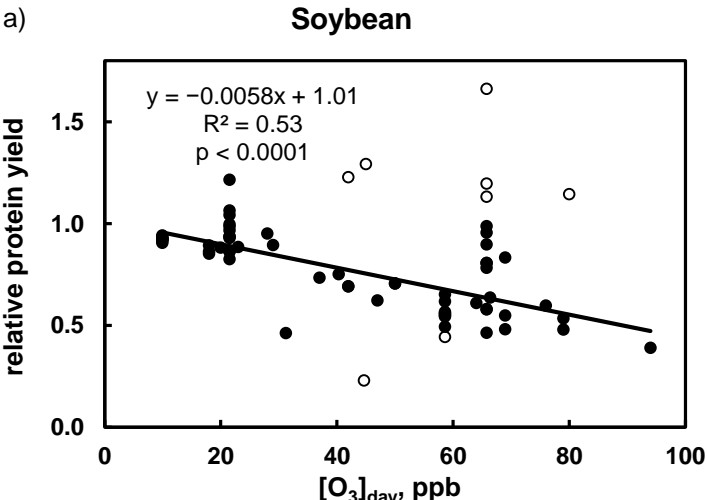

**Figure 3.** *Cont.*

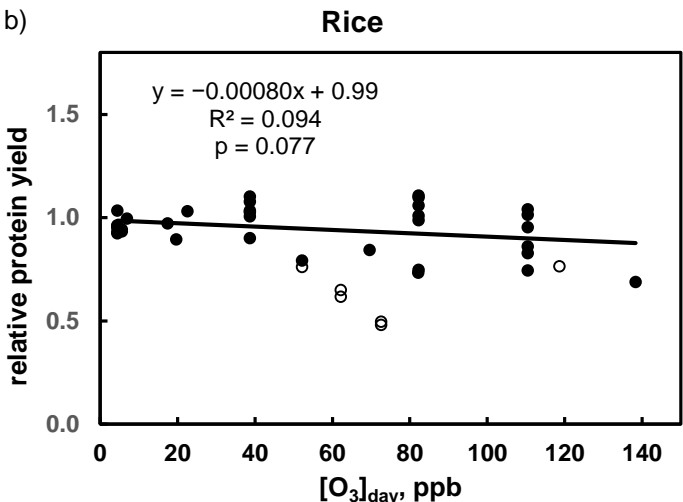

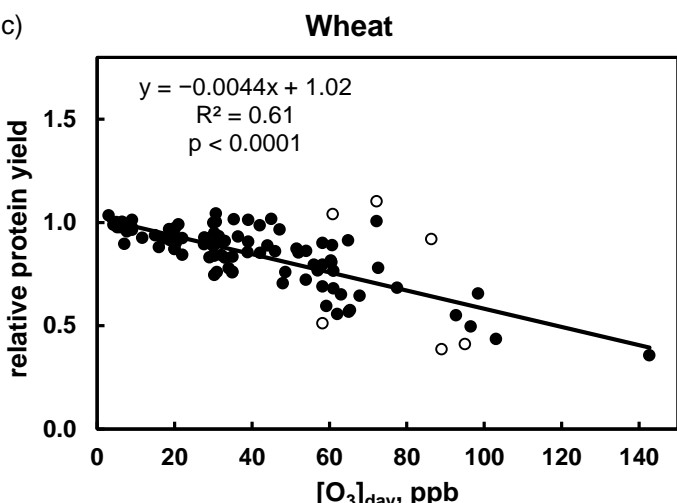

**Figure 3.** Relationship between relative seed protein yield (SPY) and daytime $O_3$ concentration, $[O_3]_{day}$, for (**a**) soybean; (**b**) rice; (**c**) wheat. Open circles denote data points representing statistical outliers for at least one of the studied response variables.

*3.4. Relationships between Relative Seed Protein Concentration (SPC) and Seed Yield (SY)*

Figure 4a–c depict the relationships between the relative SPC and relative SY, obtained from the response regression data of Figures 1 and 2, for the three crops. Rice and wheat exhibited a strongly significant negative relationship between these two response variables, which was of a similar magnitude for the two crops, although larger for rice. Soybean, on the other hand, did not show any indication of a relationship between the relative values of SPC and SY.

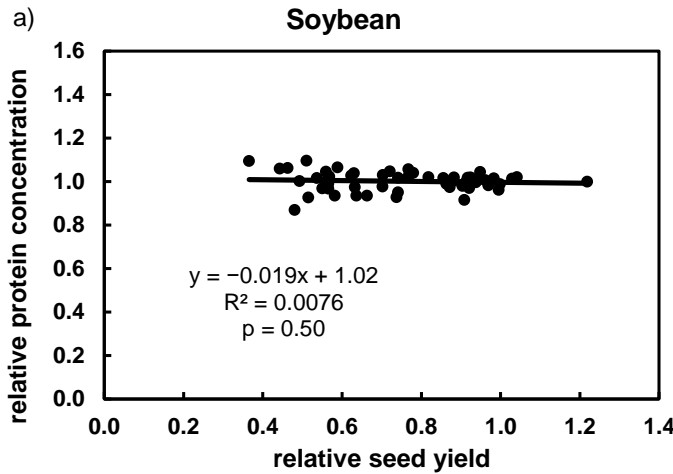

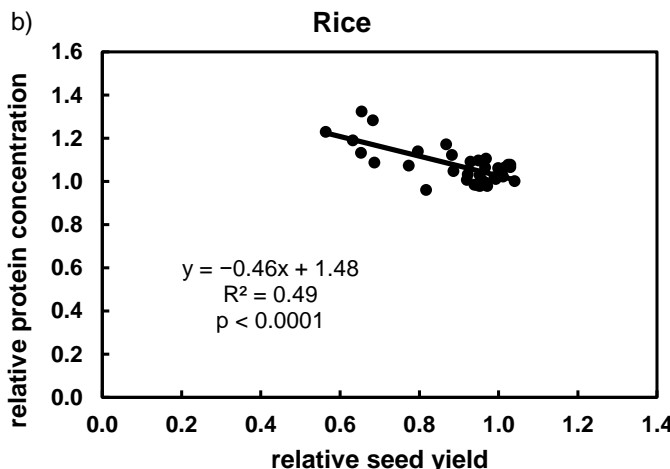

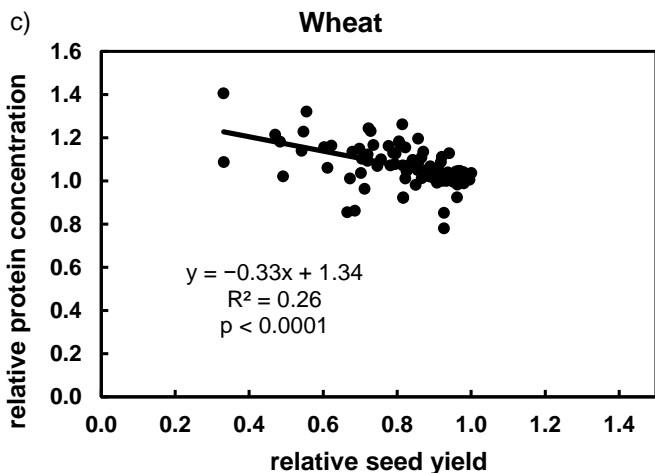

**Figure 4.** Relationship between the relative levels of seed protein concentration (SPC) and seed yield (SY) from the response regression for (**a**) soybean; (**b**) rice; (**c**) wheat (outliers excluded).

### 3.5. Assessment of the Relative Importance of O$_3$ Effects on Protein Accumulation in Soybean, Rice, and Wheat

The average protein concentration at zero O$_3$ exposure based on the database of the present study, shown in the response regression data of Figure 2a–c, was 38.7%, 9.0%, and 12.3% for soybean, rice, and wheat, respectively. Average seed yields for these three crops, according to the FAO, were (2013–2017): 2.7 ton ha$^{-1}$ (soybean), 4.6 ton ha$^{-1}$ (rice), and 3.4 ton ha$^{-1}$ (wheat).

To investigate the implications of our results, the estimated SPY based on the absolute SY from the FAO, absolute SPC from the experiments, and the relative SPY estimated from the response regressions (Figure 3) are presented in Figure 5 for the three levels of [O$_3$]$_{day}$: 10 ppb (~preindustrial), 37 ppb (present), and a hypothetical projected further elevated [O$_3$]$_{day}$ level of 60 ppb. The conclusion that the O$_3$ impact on protein production was much larger for soybean than for the other two crops would be robust even in consideration of the uncertainties around the highly simplified assumptions of the calculations behind Figure 5. The effect was larger for wheat than for rice, but this difference was much smaller than that from soybean. The improvements in SPY by reducing [O$_3$] from current to pre-industrial levels suggested by Figure 5 were 200, 10, and 70 kg protein ha$^{-1}$ for soybean, rice, and wheat, respectively.

Figure 6 shows the corresponding graph for SY. In this case, the difference in general productivity, according to FAO statistics, had a strong influence on the pattern. Still, the lower sensitivity to O$_3$ of the most productive crop, rice, as compared to wheat and soybean, is obvious. However, a certain level of relative yield loss translated into a larger absolute yield loss in a more high-yielding crop. The improvements in SY by reducing [O$_3$] from current to pre-industrial levels suggested by Figure 6 were 0.52, 0.23, and 0.54 ton ha$^{-1}$ for soybean, rice, and wheat, respectively.

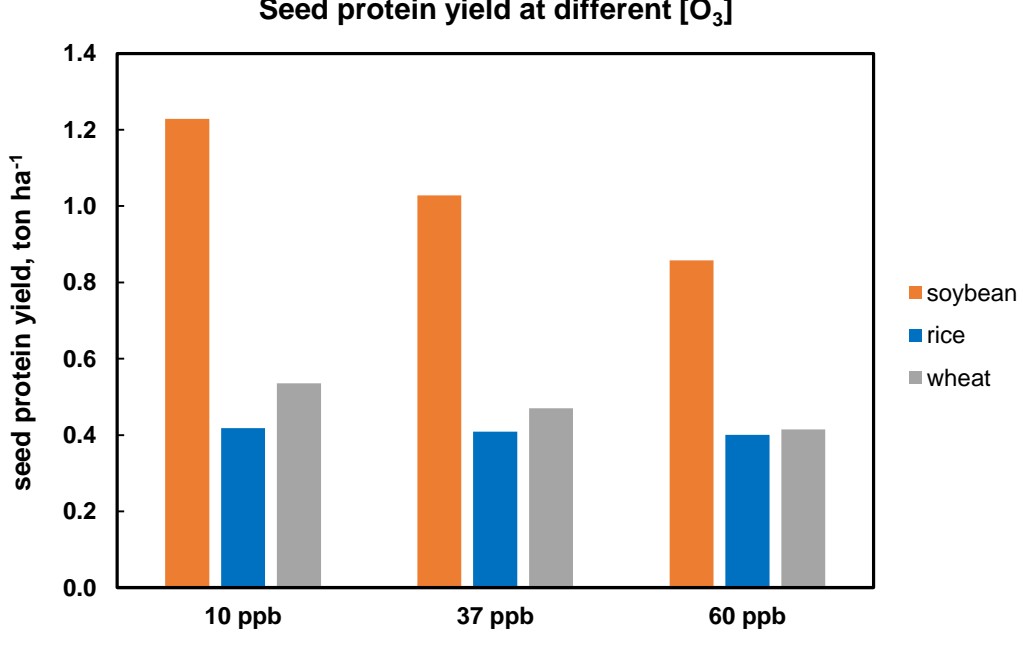

**Figure 5.** Estimation of the typical seed protein yield (SPY) based on FAO statistics for global average yields and the response functions of the present study for three O$_3$ concentration scenarios: preindustrial O$_3$ (10 ppb), the average O$_3$ levels of the control treatments in the experiments (37 ppb), and a hypothetical future increased [O$_3$] at 60 ppb.

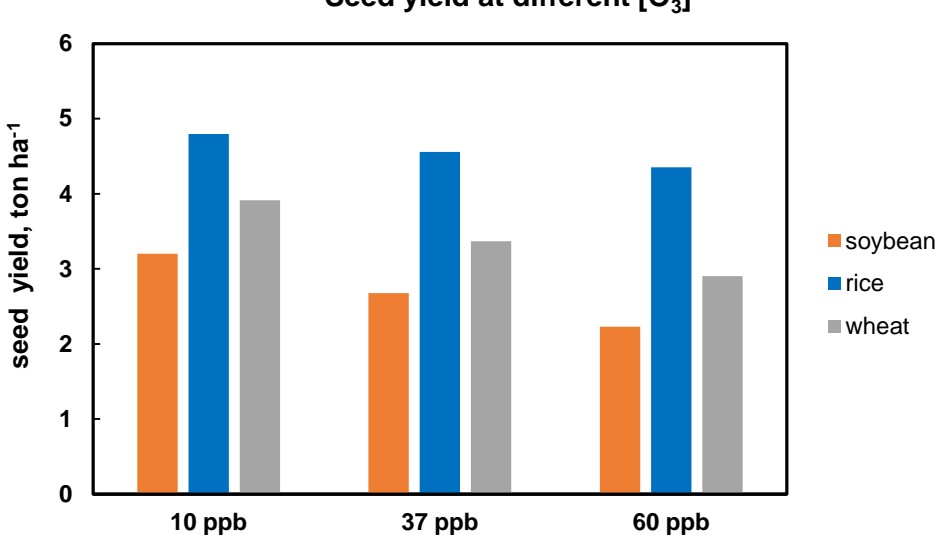

**Figure 6.** Estimation of the typical seed yield (SY) based on FAO statistics for global average yields and the response functions of the present study for three $O_3$ concentration scenarios: preindustrial $O_3$ (10 ppb), the average $O_3$ levels of the control treatments in the experiments (37 ppb), and a hypothetical future increased [$O_3$] at 60 ppb.

## 4. Discussion

Our first hypothesis that SPC is less positively affected by $O_3$ in soybean compared to wheat and rice gained support from the data analysis. Although it is not possible to draw a firm conclusion about mechanisms from a statistical analysis such as ours, it is reasonable to assume that the negative effects of $O_3$ on symbiotic N fixation in legumes observed in several studies [2,24] can explain the difference in response between soybean and the cereals obtained in our study. This conclusion is further supported by the study by Cheng et al. [25] in which $O_3$ was observed to reduce N fixation in soybean.

In line with the second hypothesis, SPY was more negatively affected by $O_3$ in soybean compared to wheat and especially to rice. This was the consequence of the combination of high sensitivity to $O_3$ with respect to SY in soybean, also highlighted in several other studies [2,23], including a larger number of experiments (here only experiments reporting protein were included), and the absence of a positive effect from $O_3$ on SPC. The stimulation of SPC by $O_3$ in wheat has been well established [4,12]. Such an effect, of the same magnitude as for wheat, was also obtained for rice in our study.

A potential source of bias in our investigation would be a change in the type of $O_3$ monitoring used since we cover data from several decades (1980–2011. We believe this, however, to be of minor importance since, with the exception of one investigation (using an $O_3$ monitor based on the chemiluminescence technique), all experiments used monitors based on the UV absorption principle, which was already well developed when the earliest experiments covered were conducted. Furthermore, the different research groups used calibration protocols to ensure accurate $O_3$ measurements.

In wheat and rice, there was a clear and strongly significant negative relationship between the $O_3$ effect on SPC and SY (Figure 4). For these crops, a loss of SY from the experimental $O_3$ treatments was significantly associated with a gain in SPC. This is in line with the yield-to-protein concentration conflict described by Kibite and Evans [32] and exemplified in many later studies. It is the so-called growth dilution effect, which has been shown to be of large importance for wheat under $O_3$ exposure [33]. In contrast to this, soybean did not show any indication of such an inverse relationship between SPC and SY. It seems that the response of SPC in soybean remains on a constant level, which is not associated with the response of SY.

The stimulation of SPC by $O_3$ in wheat and rice is likely to depend on a larger amount of N available per unit mass yield when SY is reduced. Although a weakened plant, e.g., from $O_3$ stress,

takes up less N from the soil, the shifted balance between reduced SY production and N available (the same in the different $O_3$ treatments of each experiment) still leads to an increased SPC. In soybean, the main source of N is the symbiotic N fixation, not the N available from artificial N fertilization. Instead, it becomes dependent on crop performance. The N fixation source of protein seems, under $O_3$ exposure, to be negatively affected more or less in direct proportion to the loss in SY, leaving SPC more or less constant. The SPC of a weakened soybean plant (e.g., by $O_3$), thus, cannot benefit from a larger N source in relation to SY since the N source seems to decline to the same extent as SY.

It should be noted that in wheat, the positive $O_3$ effect on SPC was substantially smaller than the negative effect on SY, resulting in a strong and significant negative effect on SPY. In the case of rice, the effects on SY and SPC were of similar magnitude and the resulting negative response of SPY was weak and non-significant. The slope coefficient in the relative effect on SPC vs. SY relationships was more negative for rice than for wheat (Figure 4). This indicates a stronger yield dilution effect on protein in rice than in wheat. However, this can, at least partly, be explained by the generally lower absolute level of SPC in rice (9.0% in our study) compared to wheat (12.3% in this study). A certain relative effect on SPC in rice represents a smaller absolute change in protein content in rice compared to wheat.

Our study of the implication of the effects of $O_3$ on seed protein accumulation (SPY, Figure 5) of the three crops at three different levels of $[O_3]_{day}$ shows that it is much more important in soybean than in the other two crops. Although the typical yield of soybean is somewhat smaller than for wheat according to FAO statistics, the protein concentration of soybean in the experiments was approximately three times higher than in wheat. This, in combination with the higher relative sensitivity of SPY in soybean (Figure 3), results in a strong sensitivity of soybean protein production to elevated $O_3$. The larger effect of $O_3$ on soybean SPY compared to wheat, and especially to rice, in line with our third hypothesis, is particularly serious considering the important role of soybean as a protein source for feed and food.

In this study, the daytime mean $O_3$ concentration, $[O_3]_{day}$, was used as the measure of $O_3$ exposure. This was the $O_3$ metric, which could be extracted for the included experiments. Ideally, one would have used the stomatal uptake of $O_3$, also known as POD (Phytotoxic Ozone Dose) which is a physiologically more relevant exposure index [13]. POD calculations require hourly data for $O_3$ and several meteorological variables in addition to further information for the experiments to accurately calculate the stomatal uptake of $O_3$. Among the crops included here, it was only available for a limited number of wheat experiments and not for soybean and rice. It should be a priority for future research on $O_3$ effects on crops to calculate POD and derive POD-based response functions for the highly important crops, soybean, and rice.

Our study shows that major crops in global food production differ considerably in $O_3$ sensitivity, not only with respect to the negative effect on SY (soybean = wheat > rice) but also regarding the stimulation of SPC (wheat = rice > soybean) and the negative effect on SPY (soybean > wheat > rice). The loss of protein in soybean cultivation is a matter with large implications, especially, which needs to be considered in analyses of food security/safety under different $O_3$ scenarios. Finally, the mechanisms behind the different responses of SPC in soybean, compared to wheat and rice, and its connection with symbiotic N fixations should be investigated further for a better mechanistic understanding of the $O_3$ sensitivity, including its agronomic implications, to the key processes of the N cycle.

## 5. Conclusions

The most important conclusions from this study were:

- The loss of SPY from $O_3$ was larger for soybean than for wheat and especially in comparison with rice.
- The different response of SPC between cereals and soybean is likely to be explained by the symbiotic N fixation symbiosis of soybean.
- A negative relationship between SPC and SY existed in rice and wheat, reflecting a larger availability of soil N per unit seed biomass when growth was hampered by $O_3$ exposure.

- In soybean, no such relationship between SPC and SY existed; this crop depends on symbiotic N for its (high) protein content and this N source seemed to decline in direct proportion to the loss of yield biomass caused by $O_3$ exposure.
- On a global scale, the adverse effects of $O_3$ on protein production are much more significant for soybean compared to the cereals.

**Supplementary Materials:** The following are available online at http://www.mdpi.com/2073-4395/10/3/357/s1, Supplementary file 1: Database for all experimental data included in this study, including full references.

**Author Contributions:** Conceptualization, H.P., M.C.B., and S.D.; methodology, M.C.B. and H.P.; investigation, M.C.B., H.P., and S.D.; data curation, M.C.B. and S.D.; writing—original draft preparation, H.P.; writing—review and editing, M.C.B.; funding acquisition, H.P. All authors have read and agreed to the published version of the manuscript.

**Funding:** This research was supported by the strategic research area BECC (Biodiversity and Ecosystem Services in a Changing Climate; http://www.cec.lu.se/research.becc).

**Acknowledgments:** Thanks for funding to the strategic research area BECC (Biodiversity and Ecosystem Services in a Changing Climate; http://www.cec.lu.se/research.becc).

**Conflicts of Interest:** The authors declare no conflict of interest.

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
