# Peer review of "Ozone Induced Loss of Seed Protein Accumulation Is Larger in Soybean than in Wheat and Rice"

_agronomy, doi:10.3390/agronomy10030357_

Round 1
Reviewer 1 Report
The manuscript titled “Ozone induced loss of protein accumulation is larger in soybean is larger in soybean than in wheat rice” by Broberg et al. concerns an interesting topic, and it would be acceptable for publishing after some minor revisions.
Keywords
Rice, soybean and wheat should be removed from keywords since already included in the title.
Title
Specify that you analyzed the SEED protein accumulation
Introduction
L29: add comma after [2,3].
L33: specification of SPY should include the SEED.
L42-43: ‘are scarce’ is in contrast with ‘several experiments’.
L52: In general, (add comma).
L56: wheat, (add comma).
L61-62: SY instead of seed yield.
L69: replace ‘high’ with ‘higher’.
Materials and methods
Did you take into account other environmental constraints overlapping with O3 exposure? These should be experiment-specific.
L98: It seems that the ROUT method is for nonlinear regression!
Results
L124: affected by increasing O3
L131: in order TO make.
L200: Y-axis of Figure 5 should start with 0.0. Graph title should be SEED protein yield…
Discussion
L212: …to draw A firm…
L218: …was more negatively affected…
Author Response
Comments and Suggestions for Authors
The manuscript titled “Ozone induced loss of protein accumulation is larger in soybean is larger in soybean than in wheat rice” by Broberg et al. concerns an interesting topic, and it would be acceptable for publishing after some minor revisions.
Keywords
Rice, soybean and wheat should be removed from keywords since already included in the title.
Response: Rice, soybean and wheat are replaced with their scientific (Latin) names in the keywords, ozone is changed to O3
Title
Specify that you analyzed the SEED protein accumulation
Response: Agreed, “seed” has been added to the title
Introduction
L29: add comma after [2,3].
Response: changed
L33: specification of SPY should include the SEED.
Response: changed
L42-43: ‘are scarce’ is in contrast with ‘several experiments’.
Response: to clarify, data on seed protein concentration (SPC) is available in several experiments, but a synthesis of O3 effects on protein accumulation (protein yield/SPY) is scarce for soybean and rice in contrast to wheat. A clarification of this has been made in the manuscript (line 44)
L52: In general, (add comma).
Response: changed
L56: wheat, (add comma).
Response: changed
L61-62: SY instead of seed yield.
Response: changed
L69: replace ‘high’ with ‘higher’.
Response: changed
Materials and methods
Did you take into account other environmental constraints overlapping with O3 exposure? These should be experiment-specific.
Response: this was not taken into account in the data analysis. We are aware that meteorological conditions have a strong influence on plant uptake of ozone, which can be taken into account by using an ozone metric based on stomatal uptake, POD (discussed at line 264-271). However, there is not available data to do this calculation for more than a few of the wheat experiments.
L98: It seems that the ROUT method is for nonlinear regression!
Response: it can also be used for linear regression (first order polynomial regression was chosen in the software (GraphPad Prism 8), which is equivalent to linear regression).
Results
L124: affected by increasing O3
Response: changed
L131: in order TO make.
Response: changed
L200: Y-axis of Figure 5 should start with 0.0. Graph title should be SEED protein yield…
Response: changed
Discussion
L212: …to draw A firm…
Response: changed
L218: …was more negatively affected…
Response: changed
Reviewer 2 Report
A reviewed manuscript contain analysis and prognosis of ground-level ozone impact of on protein accumulation in soybeans, wheat and rice. The research involved statistical analysis of literature available data. The presented results seem to be important in the light of increasing environmental pollution. The manuscript is suitable for publication after a minor revision. Please complete the discussion with an analysis of the reliability of the ground-level ozone concentration measurement results. This is a problem because research groups use different methods to measure ground-level ozone concentration during the analysed period of time (1980 and 2019). I have also problem with supplementary file (file is broken) and I'm not able to review it.
Author Response
Comments and Suggestions for Authors
A reviewed manuscript contain analysis and prognosis of ground-level ozone impact of on protein accumulation in soybeans, wheat and rice. The research involved statistical analysis of literature available data. The presented results seem to be important in the light of increasing environmental pollution. The manuscript is suitable for publication after a minor revision. Please complete the discussion with an analysis of the reliability of the ground-level ozone concentration measurement results. This is a problem because research groups use different methods to measure ground-level ozone concentration during the analyzed period of time (1980 and 2019). I have also problem with supplementary file (file is broken) and I'm not able to review it.
Response: Precision and accuracy of ozone monitors have likely improved over the time-period of the experimental data included in our analysis. Among the experiments there is only one study (Kress et al. 1983) using an ozone monitor based on the chemiluminiscence principle, which may be different fromthe remaining studies using UV absorption technique. We have added information in the methods part as well as the Discussion (lines 83-85, 230-236) on this topic.
We could not find any problem with the supplementary file, which is a Microsoft Excel Worksheet (.xlsx), but if a different file format is preferred this can be changed.